# Synthesis and Investigation of Physicochemical Properties and Biocompatibility of Phosphate–Vanadate Hydroxyapatite Co-Doped with Tb^3+^ and Sr^2+^ Ions

**DOI:** 10.3390/nano13030457

**Published:** 2023-01-23

**Authors:** Natalia Charczuk, Nicole Nowak, Rafal J. Wiglusz

**Affiliations:** 1Institute of Low Temperature and Structure Research, Polish Academy of Sciences, Okolna 2, PL-50-422 Wroclaw, Poland; 2Department of Animal Biostructure and Physiology, Wroclaw University of Environmental and Life Sciences, Norwida 25, PL-50-375 Wroclaw, Poland

**Keywords:** spectroscopy, phosphate–vanadate hydroxyapatite, Tb^3+^ and Sr^2+^ co-doping, hydrothermal method, fluorescent probe, bioimaging

## Abstract

Searching for biocompatible materials with proper luminescent properties is of fundamental importance, as they can be applied in fluorescent labeling and regenerative medicine. In this study, we obtained new phosphate–vanadate hydroxyapatites (abbr. HVps) co-doped with Sr^2+^ and Tb^3+^ ions via the hydrothermal method. We focused on examining the effect of various annealing temperatures (500, 600 and 700 °C) on the spectroscopic properties and morphology of the obtained HVps. To characterize their morphology, XRPD (X-ray powder diffraction), SEM-EDS (scanning electron microscopy–energy-dispersive spectrometry), FT-IR (Fourier transform infrared) spectroscopy and ICP-OES (inductively coupled plasma–optical emission spectrometry) techniques were used. A further study of luminescent properties and cytocompatibility showed that the obtained HVps co-doped with Sr^2+^ and Tb^3+^ ions are highly biocompatible and able to enhance the proliferation process and can therefore be potentially used as fluorescent probes or in regenerative medicine.

## 1. Introduction

Synthetic hydroxyapatites (HAps) show comparable properties to hydroxyapatites, which naturally occur in mammalian bones and teeth [1]. HAps are widely used in medicine as a bone fracture filler, dental material and tissue replacement material in compact bone [2,3]. In recent years, there has been extensive research into the use of hydroxyapatites as a matrix for drug delivery systems [4,5,6,7].

The feature that makes hydroxyapatites an extremely attractive biomaterials is the ease with which ions can be replaced in their crystal lattice [8,9]. Isomorphic substitution enables the modification of osteoconductive and osteoinductive properties, adjusts the porosity of the material and provides high mechanical strength. The substituents show high value in mineral homeostasis and the metabolic processes of the cells and tissue surrounding hydroxyapatite-based implants [10]. For instance, valuable biological properties can be gained by doping hydroxyapatite with strontium, as it plays a key role in bone mineralization [11,12]. Moreover, strontium ions stimulate bone formation by promoting the proliferation of osteoblasts and reducing bone resorption [13]. Therefore, Sr^2+^ ions stimulate the process of bone regeneration and increase bone mechanical strength [13,14,15]. Moreover, strontium is used in the form of strontium ranelate (Protelos^®^) as a drug against osteoporosis [16]. Another profitable substitution concerns the replacement of phosphate by vanadate in the anionic subnetwork of hydroxyapatite. It can result in obtaining unique acid–base catalysts and highly active oxidation catalysts due to the Lewis acidic properties and appropriate redox potential of vanadium [17]. Due to the structural similarity of vanadate to phosphate, it can take over a regulatory function in cellular processes in which phosphate plays a key role [18,19]. This phenomenon is based on vanadate’s interaction with phosphate-dependent enzymes, i.e., phosphatases and kinases [20]. In addition, the substitution of a phosphate group by a vanadate group in hydroxyapatite crystal lattice could inestimably affect its spectroscopic properties [21,22].

Furthermore, HAps doped with rare-element ions showing excellent luminescent properties can be applied in diagnostics as inorganic fluorescent probes [23,24]. Nowadays, Tb^3+^ ions are particularly widely used as dopants because of their spectroscopic properties, including intense (blue or green) luminescence, long decay times and their ability to be excited by both UV and visible radiation [25,26,27]. Biocompatibility indicates their essential role in biological applications [28,29]. Crucially, HAps doped with terbium ions can be absorbed by living cells [30,31]. As a result, fluorescent labeling involving Tb^3+^ ions enables non-invasive monitoring of the inserted implant or the drug release process.

In recent years, studies have presented incorporation of divalent and trivalent cations into the HAp structure [32,33]. Likewise, fluorapatites doped with rare-element ions were recently studied by our research group [34,35,36]. In addition, there are some studies available in the literature on apatites co-doped with various ions, including hydroxyapatites, chlorapatites, and oxyapatites, with Ln^3+^ ions and additionally built-in silicon or vanadate groups [21,37,38,39]. 

In this paper, new phosphate–vanadate hydroxyapatites (abbr. HVp) co-doped with Sr^2+^ and Tb^3+^ ions are presented. Their properties as a potential bioimaging fluorescent probe are further investigated. The structure, morphology and luminescent properties of the obtained materials are examined. Furthermore, biocompatibility tests are carried out.

## 2. Materials and Methods

### 2.1. Synthesis Method

New hydroxyapatites co-doped with Tb^3+^ and Sr^2+^ ions with up to two vanadate groups (VO_4_) substituted for phosphate groups were synthesized using the hydrothermal method. The concentration of strontium ions was fixed at 1 mol% and 2 mol%; the concentration of Tb^3+^ ions was fixed at 0.5, 1 and 2 mol%—in both cases, in a ratio of calcium ion molar content. Solutions of Ca(NO_3_)_2_∙4H_2_O (99.0–103.0% Alfa Aesar, Haverhill, MA, USA), Sr(NO_3_)_2_ (>99.0% Alfa Aesar, Haverhill, MA, USA) and (NH_4_)_2_HPO_4_ (>99.0% Acros Organics, Schwerte, Germany) were obtained by dissolving a stoichiometric amount of substrate separately in deionized water. NH_4_VO_3_ (≥99.0% ASC, Sigma-Aldrich, Saint Louis, MO, USA) solution was prepared in a microwave reactor (ERTEC MV 02-02) by mixing a stoichiometric amount of powder with deionized water in a Teflon vessel and treatment at 150 °C under a pressure of 8–11 bar for 30 min. A solution of Tb(NO_3_)_3_∙xH_2_O was prepared by digesting a stoichiometric amount of Tb_4_O_7_ (99.99% Alfa Aesar, Haverhill, MA, USA) in HNO_3_ (≥65.0%, ASC, Sigma-Aldrich, Saint Louis, MO, USA). The obtained solution was recrystallized four times to remove the excess of HNO_3_. The Ca(NO_3_)_2_∙4H_2_O, Sr(NO_3_)_2_ and Tb(NO_3_)_3_∙xH_2_O solutions were mixed, added to the previously amalgamated mixture of (NH_4_)_2_HPO_4_ and NH_4_VO_3_ and placed together into a Teflon vessel. The pH of the solution was adjusted to 9.00 by adding ammonia (NH_3_∙H_2_O 25% Avantor, Poland). The reaction was carried out in a microwave reactor (ERTEC MV 02-02) at a temperature of 240–250 °C under autogenous pressure (45–50 bar). The reaction time was set to 90 min. The final products were washed out several times with deionized water until the pH was equal to 7.00. The obtained wet powders were dried for 48 h at a temperature of 70 °C, then thermally treated at a temperatures 500, 600 and 700 °C for 3 h.

### 2.2. Structural Characterization

XRD patterns were recorded using a PANalyticalX’Pert pro x-ray diffractometer (Malvern Panalytical Ltd., Malvern, UK) equipped with Ni-filtered Cu Kα radiation (Kα_1_ = 1.5406, U = 40 kV, I = 30 mA) in a 2θ range of 5–70°. The surface morphology and elemental mapping were examined by an FEI Nova NanoSEM 230 scanning electron microscope equipped with an EDS spectrometer (EDAX PegasusXM4) operating at an acceleration voltage of 3.0 kV and spots at 2.5. An even layer of graphite was sprayed on the samples before observation. The elemental contents were specified using an inductively coupled plasma optical emission spectrometer (ICP-OES; Agilent 720, Santa Clara, CA, USA). The solutions were prepared by dissolving 50 mg of powder in 1 mL of 70% HNO_3_ (ASC, Sigma-Aldrich, Saint Louis, MO, USA) at a temperature of 120 °C and by adding deionized water to a final volume of 25 mL. The concentrations of Ca, Tb, Sr, P and V were measured using a standard setting and compared with standard curves. The concentrations of Sr, Tb, V and P atoms were measured in solutions diluted 20 times, whereas for Ca, a solution diluted 500 times was used. For the measurements, three parallel samples of the solutions were prepared; they were analyzed by comparison with standard curves in the concentration ranges of 0.05–5.00 mg/mL (for Ca, Sr, Tb and V atoms) and 10–200 mg/mL (for P atoms). The measurements of attenuated total reflectance (ATR-FT-IR) were obtained using a Nicolet iS50 infrared spectrometer (Thermo Fisher Scientific, Waltham, MA, USA). The spectra were recorded in the range of 4000–400 cm^−1^ (mid-IR) with a spectral resolution of 4 cm^−1^ (32 scans).

### 2.3. Luminescence Properties

The luminescence kinetics, emission and excitation emission spectra were obtained using an FLS980 fluorescence spectrometer (Edinburgh Instruments, Kirkton Campus, UK). All spectra were recorded at room temperature. For emission and excitation emission measurements, an 450 W Xenon Lamp was used as an excitation source in combination with a visible PMT-980 detector at a spectral resolution of 0.2 nm. The emission spectra were recorded at an excitation wavelength of 266 nm, and excitation emission spectra were recorded by monitoring the signal at 545 nm, which relates to the maximum of emission (^5^D_4_→^7^F_5_ transition) of Tb^3+^ ions [25,26,27]. Spectral correction to the detector sensitivity and excitation source intensity was applied for the emission and excitation emission measurements, respectively. A microsecond flash lamp (uF2 250 nm) and a REDPMT 500 nm mirror detector were used to record the luminescence lifetime.

### 2.4. Biological Properties

#### 2.4.1. Preparation of Sample Suspensions

The obtained powder materials were sterilized under UV light for 30 min. Then, the stocks of nanosized phosphate–vanadate hydroxyapatite co-doped with Tb^3+^ and Sr^2+^ ions were prepared by the suspension of compounds in sterile, distilled water at a concentration of 1 mg/mL. Subsequently, an ultrasonic bath was used to agitate particles in each sample of the colloidal solution (for 1 h at RT). Freshly prepared colloids were additionally sterilized under UV light for 30 min for use in biological experiments.

#### 2.4.2. Cell Culture and Viability Assay

Normal dermal human fibroblast (NHDF–C-12300, Lonza, Basel, Switzerland) and mouse fibroblast (L929, CCL-1, ATCC, Manassas, VA, USA) cell lines were maintained in high-glucose DMEM (4.5 g/mL) without phenol red (Biological Industries, Beit-Haemek, Israel), which was supplemented with 10% FBS (Biological Industries, Beit-Haemek, Israel), 2 mM L-glutamine and 25 μg/mL gentamicin (Biological Industries, Beit-Haemek, Israel). Cell lines were maintained under the standard conditions of 5% CO_2_ in a humified atmosphere of 95% air and 5% CO_2_ at 37 °C. NHDF and L929 cells were subcultured once a week with TrypLE solution (Thermo scientific, Waltham, MA, USA, TrypLE™ Express Enzyme (1X), no phenol red, Thermo Fisher Scientific, Waltham, MA, USA). The cell lines were passaged three times before the experiments were conducted. Moreover, all plastic materials such as serological pipettes, pipette tips, centrifuge falcon tubes, plastic culture flasks and 96-well culture plates were purchased from SPL Life Sciences (Pochon, Republic of Korea).

To evaluate the biocompatibility of the obtained compounds, an MTT viability assay was performed on NHDF and L929. Both types of cells were seeded at a density of 1 × 10^4^ cells per well in a 96-well plate; then, the cells were treated with two series of compounds at three different concentrations: 10 µg/mL, 50 µg/mL and 100 µg/mL. The incubation time was established as 24 h. After that time, the cells were washed out once with sterile PBS. Then, MTT (Sigma-Aldrich, Saint Louis, MO, USA) solution (0.5 mg/mL) dissolved in sterile PBS was added to the cells. The untreated cells were established as a negative control and for this group, and cell viability was established at 100%. The experiment was carried out under standard conditions of 5% CO_2_, 37 °C and 95% humidity. After 3 h of incubation, the MTT solution was removed, and formazan crystals were dissolved with isopropanol. Absorbance was read at 570 nm. The experiment was conducted three times. Cell viability was estimated using the following formula: $$\mathrm{Cell}\;\mathrm{viability}=\frac{\mathrm{sample}\;\mathrm{absorbance}}{\mathrm{control}\;\mathrm{absorbance}}\cdot 100$$

## 3. Results and Discussion

### 3.1. Characterization of Structure and Morphology

To investigate the crystalline structure of the obtained materials, all the recorded XRD patterns were compared with the reference hydroxyapatite pattern from the Inorganic Crystal Structure Database (ICSD 253617). The hexagonal structure, which is ascribed to the P6_3_/m space group of the hydroxyapatites (Figure 1), was confirmed for all compounds [40,41]. However, the peak originating from β-strontium hydrogen phosphate (SrHPO_4_, β-SHP) is visible (Figure 2a–d) but does not affect the HVp properties. [42,43]. In the case of the obtained materials, isomorphic substitution takes place in both the cationic and anionic subnetwork. In the cationic subnetwork, Ca^2+^ ions, the ionic radius of which equals 0.99 Å, are partially replaced by Sr^2+^ ions (1.13 Å) and Tb^3+^ ions (0.923 Å) [44,45]. These Ca^2+^ ions represent two crystallographically independent calcium atoms: Ca(I) and Ca(II). Ca(I) is located at 6h [xy1 = 4], with the local symmetry m, while Ca(II) is located at the position 4f [1/3 2/3 z], with the local symmetry 4f [46]. Despite minor differences in the values of ionic radii and low dopant concentrations (1 and 2 mol% for Sr^2+^ ions and 0.5, 1 and 2 mol% for Tb^3+^ ions), disturbances in the crystal lattice occurred and manifested as a slight shift in 2θ values and changes in half-width of the peaks [43].

Furthermore, substitution of Ca^2+^ ions by Tb^3+^ ions is associated with a lack of charge compensation. The failure to equalize the charge within substitution can cause spontaneous compensation (for example, by forming an O^2−^ ion as the second oxygen form next to the OH^−^ ion), the consequences of which are changes in unit cell parameters [25]. In the case of the anionic subnetwork, there is an isomorphic exchange between the PO_4_^3−^ (1.10 Å) and VO_4_^3−^ (1.22 Å) ions [47]. The incorporation of larger VO_4_^3−^ ions into the HVp crystal lattice results in an increase in the value of lattice parameters and the lattice volume [48]. For the standard hydroxyapatite diffraction pattern (ICSD 253617), peaks in the range of 31–35° mostly originate from the phosphate group [49,50]. For the obtained diffractograms, in comparison to the standard ICSD pattern (Figure 2a–d), the half-width of the mentioned peaks increased. The widening of peaks demonstrates the results of doping the hydroxyapatite with up to two vanadate groups [51].

In the current study, we investigated the impact of various annealing temperatures on the materials. An increase in the temperature in the range of 500–700 °C is correlated with an increase in crystallinity [52,53]. As the annealing temperature increases, the half-width of the peaks decreases (Figure 2a–d). For the thermally untreated samples (as prepared) and the samples annealed at 500 and 600 °C, a single crystallographic phase assigned to the structure of hydroxyapatite is present (with an insignificant portion of the phase derived from β-SHP). However, for the materials annealed at 700 °C, peaks originating from Ca_2_V_2_O_7_ (calcium pyrovanadate) and monetite, i.e., anhydrous dicalcium phosphate, are present in the range of 27–32° (Figure 2d) [54,55]. This observation indicates that a temperature up to 600 °C is suitable for annealing phosphate–vanadate hydroxyapatite co-doped with Sr^2+^ and Tb^3+^ ions.

The results of scanning electron microcopy show that the particles tend to agglomerate (Figure 3a). Agglomeration is a natural tendency within HVps synthesized via the hydrothermal method in aqueous solvent [56]. SEM mapping confirms the presence of selected elements, i.e., oxygen, calcium, terbium, strontium, phosphorus and vanadium (Figure 3b). The ICP-OES results are in agreement with the theoretically calculated values (Table 1). For materials doped with 0.5 and 1 mol% Tb^3+^ ions, one phosphate group was substituted by vanadate; however, for materials doped with 2 mol% Tb^3+^ ions, two vanadate groups were incorporated in the hydroxyapatite framework.

In all recorded ATR-FT-IR spectra (Figure 4a–c), bands originating from the phosphate (PO_4_^3−^), vanadate (VO_4_^3−^) and hydroxyl (OH^−^) groups are present. Their locations are consistent with the data available in the literature and confirm that the materials have the hexagonal structure of phosphate–vanadate hydroxyapatite [17,19,48,57,58].

In the case of the spectra (Figure 4a) measured for the material co-doped with 1 mol% Tb^3+^ and 1 mol% Sr^2+^ annealed at 600 °C, the bands characteristic of the phosphate group are located at 560, 600, 960, 1024 and 1086 cm^−1^ [17]. The bands at 560 cm^−1^ and 600 cm^−1^ can be assigned to the triple-degenerated stretching mode, ν_4_(PO_4_^3−^). The band at 960 cm^−1^ originates from the non-degenerated symmetric stretching mode, ν_1_(PO_4_^3−^) [59]. The 1024 cm^−1^ and 1086 cm^−1^ bands correspond to the triple-degenerated antisymmetric stretching mode, ν_3_(PO_4_^3−^) [59]. The vanadate group is represented by the bands at 849, 870 and 887 cm^−1^, which originate from the symmetrical stretching mode, ν_3_(VO_4_^3−^) [60,61]. Additionally, a band with an assignment that is not straightforward appears at 469cm^−1^. In the 400–500 cm^−1^ range, bands can be attributed to modes originating from both phosphate and vanadate groups. The reason for this is an overlap of bands originating from ν_2_(PO_4_^3−^), ν_2_(VO_4_^3−^) and ν_4_(VO_4_^3−^) modes [19]. The presence of the hydroxyl group is confirmed by the presence of the bands at 630 and 3567 cm^−1^ [62,63]. A band originating from the symmetrical stretching mode of the hydroxyl group is present at 3567 cm^−1^, whereas the band originating from the torsion mode of the hydroxyl group is present at 630 cm^−1^. The torsion mode corresponds to the rotation of a hydrogen atom around the C axis in the apatite crystal lattice (libration band) [64]. Both bands are characterized by relatively low intensity. For the material not thermally treated (as prepared) and materials annealed at 500, 600 and 700 °C, the positions of the bands do not differ significantly (Table 2). This feature implies that the temperature does not have a major effect on wavenumber shifts.

Band shifts and intensity changes can be observed in the ATR-FT-IR spectra recorded for both thermally untreated samples and samples annealed at 600 °C (Figure 4b,c). According to the results of the ICP-OES analysis (Table 1), only in the case of the materials doped with 2 mol% Tb^3+^, two vanadate groups were incorporated into the hydroxyapatite crystal lattice. In the ATR-FT-IR spectra recorded for those samples, the intensity of the bands with origins assigned to the vanadate group is higher than for the other materials, in which only one vanadate group is present. The intensity increment for the bands originating from vanadate stretching modes (ν_3_) correlates with a decrease in the intensity of the phosphate stretching modes (ν_3_) (Figure 4b,c). Furthermore, the spectra show the shift (~5 cm^−1^) of the bands originating from the vanadate stretching modes (ν_3_(VO_4_^3−^)) towards the lower wavenumber values. The change in the position and intensity of the bands is related to their widening, which implies a lower level of crystallinity of compounds with two built-in vanadate groups [60,61]. Moreover, there are reports in the literature on shifts in the selected bands caused by changes in the content of vanadate groups [17,48]. An example of this phenomenon is the shift of ν_s_(OH^−^) towards lower wavenumber values with a simultaneous increase in the content of vanadates [17,48]. Nevertheless, in the case of the analyzed compounds, there is an insignificant difference in the position of this band (the band is located at 3566 cm^−1^ for materials with one vanadate group and at 3565 cm^−1^ for materials with two vanadate groups). 

### 3.2. Investigation of Luminescence Properties

The examination of luminescent properties of the obtained materials confirms the presence of Tb^3+^ ions. Good-quality spectra were recorded for phosphate–vanadate hydroxyapatite co-doped with Sr^2+^ and Tb^3+^ ions, for which terbium concentrations were fixed at 0.5, 1 and 2 mol% in a ratio of calcium ion molar content. Emission spectra were recorded upon excitation at 266 nm, and excitation emission spectra were recorded by monitoring the signal at 545 nm, which relates to the maximum of emission (^5^D_4_→^7^F_5_ transition) of Tb^3+^ ions [25,26,27].

In the excitation emission spectra of the obtained materials, the peaks characteristic of Tb^3+^ ions are present (Figure 5). The origin of the peaks in the near-UV energy range is related to the possibility of the occurrence of interconfigurational and charge–transfer transitions, as well as defects present in the hydroxyapatite crystal lattice, resulting in the absorption of radiation by the matrix, which, in this case, plays the role of a sensitizer (or, more likely, the appearance of a self-activated HAp fluorescence) [65,66]. The occurrence of self-activating HAp fluorescence is caused by (i) crystal defects, such as Ca^2+^, O^2−^ and OH^−^ vacancies; (ii) isomorphic substitution, as well as the presence of impurities; (iii) asymmetry of the structure caused by the changes in the length and the angle of Ca-O and P-O bonds [67,68,69]. These factors are induced by the thermal treatment of the material and the manipulation of the pH during synthesis. Moreover, the exposure of the material to high-energy radiation increases the probability of excitation of the HAp matrix. The absorbed radiation can then be transferred to the activator ion (in this case, the Tb^3+^ ion), which allows emission to be observed. Nonetheless, in the case of the analyzed materials, CT within the vanadate group (O^2−^→V^5+^) and/or the interconfigurational transitions (4f^8^→4f^7^5d^1^) of Tb^3+^ ions are more likely to be responsible for the presence of the discussed peaks. These transitions are allowed by the selection rules, which explains the wide half-width of the peaks and their high intensity. However, the bands present in the indicated range are overlapping, which impedes a straightforward distinction of their origin. As previously established, the host matrix has a strong influence on the position of the peaks originating from the 4f^8^→4f^7^5d^1^ transitions of Tb^3+^ ions [70]. In the case of the obtained materials, the band at λ = 250 nm may originate from the transition of the electron to the low-spin 5d^1^ state. In addition, a band of low intensity can be observed in the spectra located at λ = 317 nm. Its origin is related to the forbidden transition from the 4f^8^ orbital to the high-spin 5d^1^ state. Changes in the strength and symmetry of the local crystal field surrounding the activator ions, which are implied by its chemical environment, significantly affect the selection rules for the electronic transitions of Tb^3+^ ions, which enables their relaxation and, as a consequence, the occurrence of such a band in the spectra. Charge–transfer transition can also take place within the hydroxyapatite matrix in the vanadate group (O^2−^→V^5+^; the electron is transferred from the 2p orbital of oxygen to the 3d orbital of vanadium). The peaks originating from O^2−^→V^5+^ CT should occur in higher wavenumbers than 4f^8^→4f^7^5d^1^ transitions of Tb^3+^ ions because less energy is required to observe the CT VO_4_ transitions (^1^A_1_ → ^1^T_1_, ^2^T_2_) (the energy gap between the terms is smaller than in the case of f-d transitions) [71]. Nevertheless, the positions of the peaks are strongly dependent on the type and nature of the ligands surrounding the V^5+^ ion and on the interaction of ions in the crystal lattice [22]. When analyzing the position of the CT transition bands, as in the case of interconfigurational transitions, it should be taken into account that this parameter is influenced by many factors, including the local environment of ions and the nature of their mutual interactions in a crystal lattice. Moreover, Tb^3+^ ions may occupy more than one crystallographic position in the HVp crystal lattice and may also show several types of interactions with other ions.

In the visible radiation range of the spectra, there are peaks related to forbidden 4f-4f intraconfigurational transitions of Tb^3+^ ions: ^7^F_6_→^5^G_4_ (353 nm), ^7^F_6_→^5^G_5_ (358 nm), ^7^F_6_→^5^L_10_ (370 nm), ^7^F_6_→^5^G_6_ (376 nm), ^7^F_6_→^5^D_3_ (380 nm) and ^7^F_6_→^5^D_4_ (485 nm). Moreover, the peak originating from the ^7^F_6_→^5^D_4_ transition is split up into five Stark components. Furthermore, in the examined emission excitation spectra, there is a region of reabsorption (λ = 450 nm and 472 nm) connected to the presence of vanadate groups in the hydroxyapatite. Molecular orbitals 3d of V^5+^ ions have T_d_ symmetry and are described as the crystal field levels: the ground state ^1^A_1_ and the excited states ^1^T_1_, ^1^T_2_, ^3^T_1_ and ^3^T_2_ [72]. For T_d_ symmetry, according to the spin selection rule, the transfer of the electron from the ground state level to the excited states ^1^T_1_ and ^1^T_2_ is allowed, whereas the transitions ^1^A_1_ → ^3^T_1_, ^3^T_2_ and emission (^3^T_1_, ^3^T_2_ → ^1^A_1_) are forbidden. However, when the ideal symmetry is disturbed, it is possible for forbidden processes to take place [22]. During excitation, the charge is transferred from the nondegenerate ^1^A_1_ state to the triple-degenerate excited states (^x^T_y_, where x = 1 or 3 and y = 1 or 2). Then, during emission, the energy is transferred radially to the ground-state level, which may result in the appearance of peaks in the indicated range in the spectra.

Tb^3+^ ions show strong blue (in the range of 370–480 nm; ^5^D_3_→^7^F_J_ transitions) or green (in the range of 470–720 nm; ^5^D_4_→^7^F_J_ transitions) luminescence [26]. The obtained emission spectra (Figure 6) show four peaks characteristic of Tb^3+^ ions originating from the ^5^D_4_→^7^F_J_ transitions (J = 3,4,5,6) [73,74,75]. Because the values of J for Tb^3+^ ions satisfy the 2J + 1 rule, the levels undergo Stark splitting and are therefore split into sublevels under the influence of the crystal field, causing peaks to be divided into components. This suggests that the terbium ions occupy different crystallographic positions, in addition to highlighting the importance of the influence of the host matrix on its spectroscopic properties [76]. As shown in Figure 6, the number of Stark components varies in the spectra depending on the annealing temperature. For thermally untreated materials (as prepared), as well as the series annealed at 500 and 600 °C, the spectra are similar. The first peak is split into three components at 485, 490 and 495 nm and corresponds to the ^5^D_4_→^7^F_6_ transition. The peak of the highest intensity (^5^D_4_→^7^F_5_ transition) responsible for the green luminescence of Tb^3+^ ions appears in the spectra at 545 nm [77]. This peak consists of two incompletely separated components. The peak originating from the ^5^D_4_→^7^F_4_ transition has two Stark components at 583 and 590 nm. The ^5^D_4_→^7^F_3_ transition is responsible for the appearance of a peak split into three components (λ = 615, 619 and 623 nm) in the case of a series of materials not subjected to heat treatment (as prepared) and annealed at 500 °C and for the appearance of one component in the case of the series annealed at 600 °C (λ = 622 nm). However, the material co-doped with 1 mol% Tb^3+^ and 2 mol% Sr^2+^ differs from the described dependency, as its peaks originating from ^5^D_4_→^7^F_6_ and ^5^D_4_→^7^F_5_ transitions split into two and one component, respectively. In the case of the series annealed at 700 °C, the spectra differ from the spectra recorded for the other series; moreover, they differ within the series. Within the series, as the concentration of Tb^3+^ ions increases, the number of Stark components increases simultaneously. Differences occur between materials doped with 0.5 or 1 mol% and materials doped with 2 mol% of Tb^3+^ ions. For the materials doped with 0.5 or 1 mol%, the peak originating from ^5^D_4_→^7^F_6_ splits into one (λ = 490 nm) or two components (λ = 485 nm i 490 nm), whereas the peaks originating from ^5^D_4_→^7^F_3_ do not split (λ = 622 nm). For materials doped with 2mol% Tb^3+^ ions, these peaks split into two components (λ = 485 i 490 nm) for Ca_9.7_Sr_0.1_Tb_0.2_(PO_4_)_4_(VO_4_)_2_(OH)_2_ or four components (λ = 485, 488, 491, 496 nm) for Ca_9.6_Sr_0.2_Tb_0.2_(PO_4_)_4_(VO_4_)_2_(OH)_2_ (^5^D_4_→^7^F_6_ transition) and three components (λ = 615, 619, 623 nm) for peaks originating from the ^5^D_4_→^7^F_3_ transition. Additionally, for the material co-doped with 2 mol% Tb^3+^ and 2 mol% Sr^2+^, the peak originating from the ^5^D_4_→^7^F_5_ transition splits into three components (λ = 543, 546, 549 nm), whereas for the other compounds, there are only two components (λ = 543 i 546 nm). Nonetheless, the characteristic of the peak originating from the ^5^D_4_→^7^F_4_ transition is consistent within all the series. In general, for the series annealed at 700 °C, the number of Stark components for ^5^D_4_→^7^F_6_ and ^5^D_4_→^7^F_3_ transitions is smaller than for the other series (except for the mentioned HVp co-doped with 2 mol% Tb^3+^ and 2 mol% Sr^2+^). In all probability, the annealing temperature could be responsible for this phenomenon. Because the materials annealed at 700 °C do not contain a single phase, their crystallographic properties and the impact of their crystal field on terbium ions may fluctuate, which directly influences the formation of Stark components.

Figure 7 shows the decrease in fluorescence intensity (proportional to the quantity of excited molecules) as a function of time measured in milliseconds. In the case of the obtained materials, fluorescence decay is not monoexponential. This may suggest that Tb^3+^ ions occupy more than one position in the crystal lattice of HVps [78]. Furthermore, for all materials, the average decay time was determined to range from 1.41 to 2.77 ms (Table 3). No dependence of the annealing temperature on the fluorescence decay time was observed. Moreover, the increase in the value of vanadate groups incorporated into the HVps did not cause the quenching of the fluorescence intensity of Tb^3+^ ions.

### 3.3. Evaluation of Biological Properties

#### Biocompatibility of Obtained Compounds

The results clearly indicate that the obtained compounds exhibit biocompatible properties toward the NHDF and L929 cell lines (Figure 8). The cell viability of the L929 cell line after 24 h of incubation with various concentrations of the tested materials is maintained above 100% compared with the control group. The L929 cell line preferred less concentrated colloidal solutions (10 and 50 ug/mL) than more concentrated solutions. Nevertheless, when cells were treated with the most highly concentrated solution, cell viability was still maintained around 120%, especially for compounds containing dopant concentration of 0.5 mol% Tb^3+^ and 1 mol% Sr^2+^, 1 mol% Tb^3+^ and 1 mol% Sr^2+^, 2 mol% Tb^3+^ and 1 mol% Sr^2^, and 2 mol% Tb^3+^ and 2 mol% Sr^2+^. However, one exception was observed when L929 cells were treated with HVp co-doped with 0.5 mol% Tb^3+^ and 2 mol% Sr^2+^. The more concentrated the colloidal solution, the fewer viable cells were observed; however, compared with the control group, cell viability was maintained at 100% or more. A slightly different situation was observed for the NDHF cell line. The repetitive tendency of a low cell proliferation rate was observed when cells were treated with the lowest concentration (10 ug/mL), especially for the sample co-doped with 0.5 mol% Tb^3+^ and 1 mol% Sr^2+^ and the sample co-doped with 2 mol% Tb^3+^ and 2 mol% Sr^2+^. However, the more concentrated the colloidal solution, the more viable cells were observed. This trend was observed among all tested colloidal solutions of phosphate–vanadate hydroxyapatite co-doped with Tb^3+^ and Sr^2+^ ions. Furthermore, excellent results were achieved when the NHDF cell line was treated with the highest tested concentration of 100 ug/mL. The results for this cell line indicate that the obtained compounds are highly compatible and able to enhance the proliferation process. In conclusion, all tested compounds showed biocompatible properties toward normal cell lines NDHF and L292. The results indicate that the obtained compounds can be further investigated as bioimaging probes, even for in vivo tests.

## 4. Conclusions

New compounds with the structure of phosphate–vanadate hydroxyapatite co-doped with Sr^2+^ and Tb^3+^ ions were obtained by a hydrothermal method. The dopant concentration was set at 1 or 2 mol% for Sr^2+^ ions and 0.5, 1 or 2 mol% for Tb^3+^ ions and confirmed by the ICP-OES technique. Furthermore, XRD analysis confirmed the hexagonal structure of hydroxyapatite (*P6_3_/m* group). As a result of increasing annealing temperature, an increase in the degree of crystallinity of the studied materials was observed (comparing the series not subjected to heat treatment with those annealed at 500, 600 and 700 °C). Moreover, annealing at 700 °C caused the formation of additional chemical compounds. In this case, a temperature of 600 °C is optimal for heat treatment. The presence of PO_4_^3−^, VO_4_^3−^ and OH^−^ groups was determined by ATR-FT-IR analysis. The characteristics of the bands originating from the *ν*(VO_4_^3−^) modes were influenced by an increase in the number of built-in vanadate groups. The obtained materials showed spectroscopic properties characteristic of Tb^3+^ ions. In the case of emission excitation spectra, a peak related to the charge–transfer transition of the vanadate group (O^2−^→V^5+^) and/or the *4f^8^→4f7^5^d^1^* interconfigurational transition of Tb^3+^ ions was present. In addition, low-intensity peaks from *f-f* Tb^3+^ intraconfigurational transitions were observed. The emission spectra showed four peaks originating from ^5^D_4_→^7^F_J_ transitions, where J = 3,4,5,6. The decay of fluorescence in the case of the studied materials was not monoexponential, and the lifetimes were in the range of 1.41–2.77 ms. All obtained materials (series annealed at 600 °C) showed a positive effect on L929 and NDHF cells and enhanced the process of proliferation. Moreover, the tested materials were biocompatible with the selected L929 and NDHF cell lines. The best biological properties for both cell lines were obtained for the materials doped with 2 mol% Tb^3+^ ions with two built-in vanadate groups.

In conclusion, the obtained phosphate–vanadate hydroxyapatite material co-doped with Sr^2+^ and Tb^3+^ ions are expected to be evaluated in future investigations as bioimaging probes, even for in vivo tests. Overall, the presented materials show promising physicochemical properties for potential bioimaging applications.

## Figures and Tables

**Figure 1 nanomaterials-13-00457-f001:**
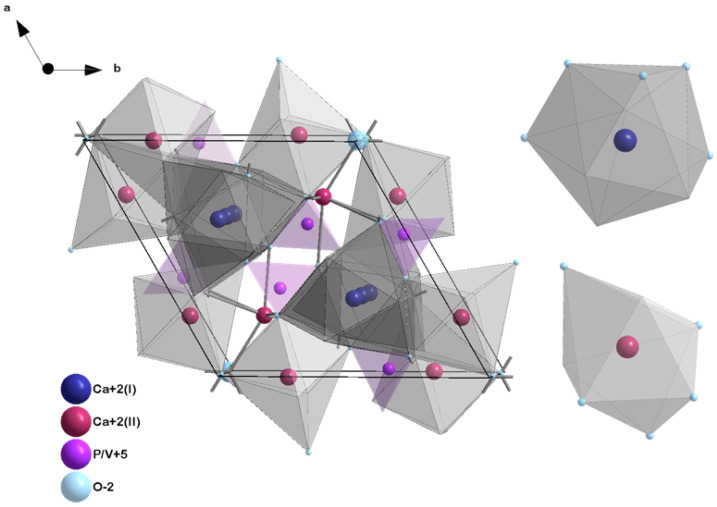
Unit cell of HVp structure. Ca(I) and Ca(II) sites are distinguished, and both Sr^2+^ and Tb^3+^ ions can be located during substitution.

**Figure 2 nanomaterials-13-00457-f002:**
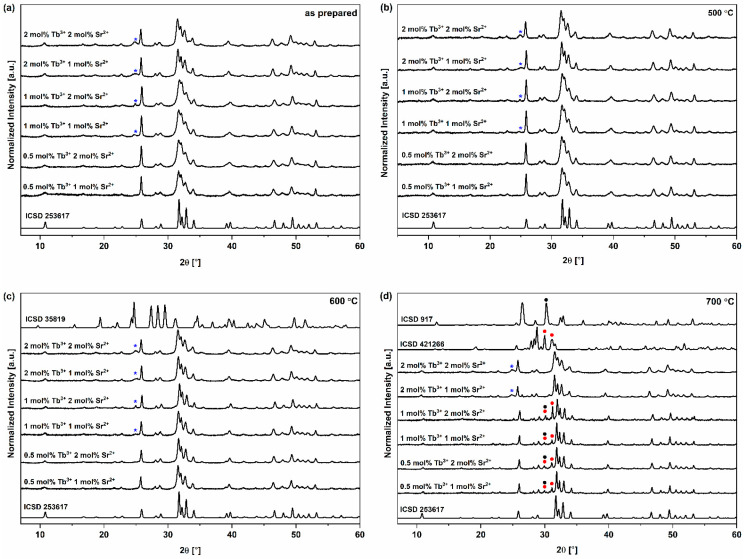
XRD results for the obtained materials: (**a**) not submitted to thermal treatment (as prepared). (**b**) annealed at 500 °C. (**c**) annealed at 600 °C. (**d**) annealed at 700 °C. The hexagonal structure, which is ascribed to the P6_3_/m space group of the HVps, was confirmed for all compounds (**a**–**d**). ICSD 253617 is database hydroxyapatite pattern. (*****) indicates peaks originating from SrHPO_4_ (ICSD 35819). For the materials annealed at 700 °C (**d**) peaks originating from CaHPO_4_ (●, ICSD 917) and Ca_2_V_2_O_7_ (●, ICSD 421266) occur in the range of 26–32°.

**Figure 3 nanomaterials-13-00457-f003:**
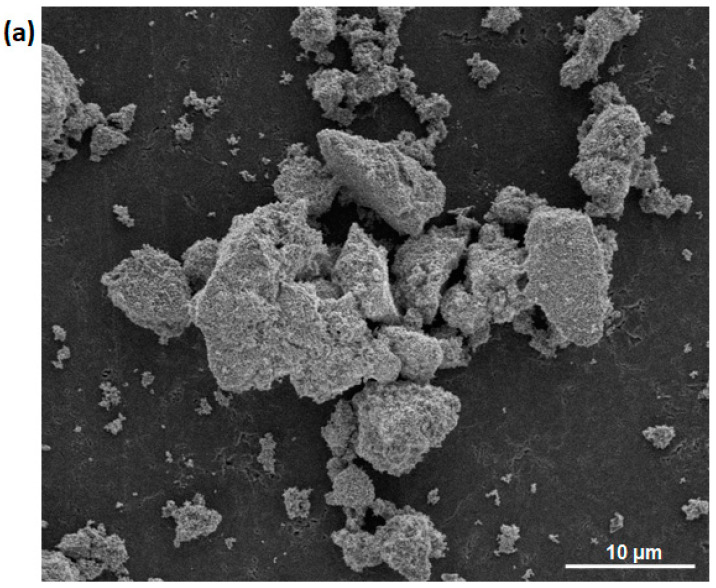
(**a**) SEM image of HVp co-doped with 1 mol% Sr^2+^ and 1 mol% Tb^3+^ annealed at 700 °C. (**b**) SEM-EDS elemental mapping of HVp co-doped with 1 mol% Sr^2+^ and 1 mol% Tb^3+^ ions annealed at 700 °C. Images show elemental composition of the sample, which contains calcium, strontium, terbium, phosphate, vanadium and oxygen elements.

**Figure 4 nanomaterials-13-00457-f004:**
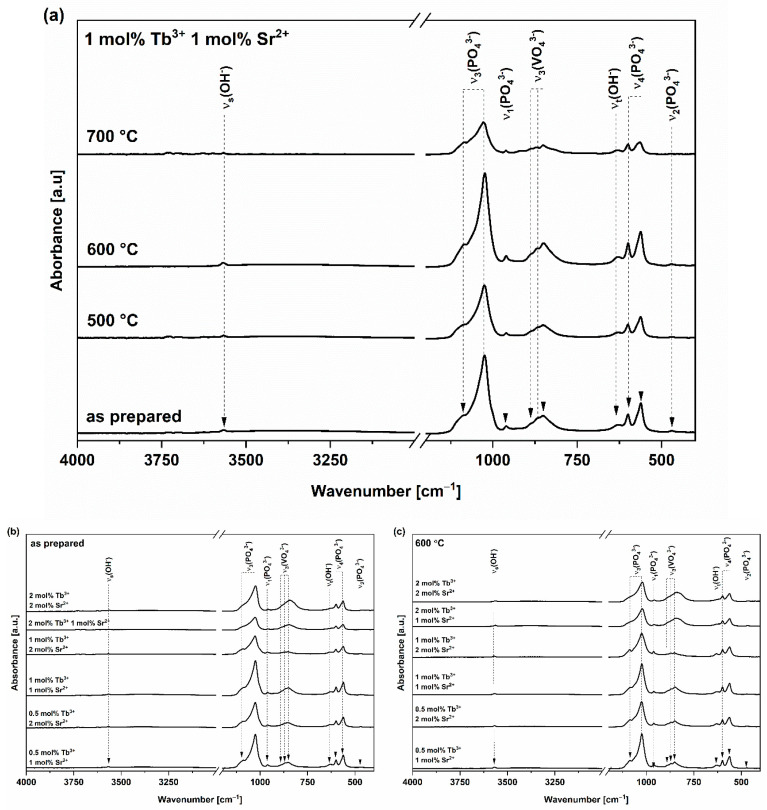
(**a**) ATR-FT-IR spectra of HVp co-doped with 1 mol% Sr^2+^ and 1 mol% Tb^3+^ ions: sample thermally untreated (as prepared) and annealed at 500, 600 and 700 °C. (**b**,**c**) ATR-FT-IR spectra recorded for (**b**) thermally untreated materials (as prepared) and (**c**) materials annealed at 600 °C.

**Figure 5 nanomaterials-13-00457-f005:**
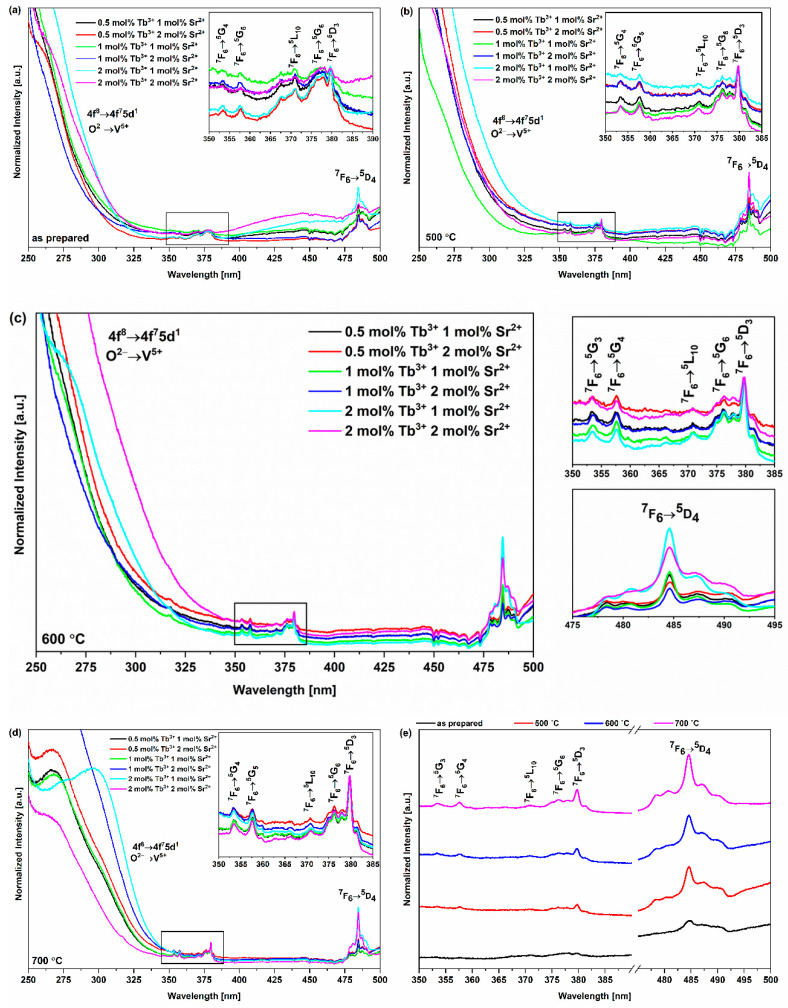
Excitation emission spectra of (**a**) thermally untreated materials (as prepared), (**b**) materials annealed at 500 °C, (**c**) materials annealed at 600 °C (**d**) materials annealed at 700 °C and (**e**) material co-doped with 2 mol% Sr^2+^ and 2 mol% Tb^3+^, at an observation wavelength of 545 nm (^5^D_4_→^7^F_5_ transition). The peak visible in the near-UV region originates from CT within the vanadate group (O^2−^→V^5+^) and/or interconfigurational transitions (4f^8^→4f^7^5d^1^) of Tb^3+^ ions. The peaks originating from 4f-4f intraconfigurational transitions of Tb^3+^ ions represent ^7^F_6_→^5^G_4_ (353 nm), ^7^F_6_→^5^G_5_ (358 nm), ^7^F_6_→^5^L_10_ (370 nm), ^7^F_6_→^5^G_6_ (376 nm), ^7^F_6_→^5^D_3_ (380 nm) and ^7^F_6_→^5^D_4_ (485 nm) transitions.

**Figure 6 nanomaterials-13-00457-f006:**
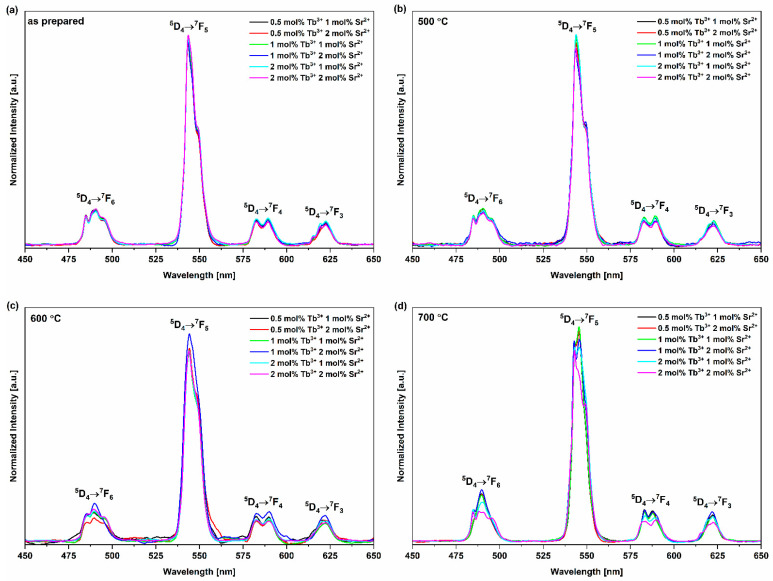
Emission spectra of series (**a**) thermally untreated (as prepared), (**b**) annealed at 500 °C, (**c**) annealed at 600 °C and (**d**) annealed at 700 °C at an excitation wavelength of 266 nm. There are four peaks characteristic of Tb^3+^ ions originating from 4f-4f intraconfigurational ^5^D_4_→^7^F_J_ (J=3,4,5,6) transitions. The number of Stark components varies in the spectra depending on the annealing temperature.

**Figure 7 nanomaterials-13-00457-f007:**
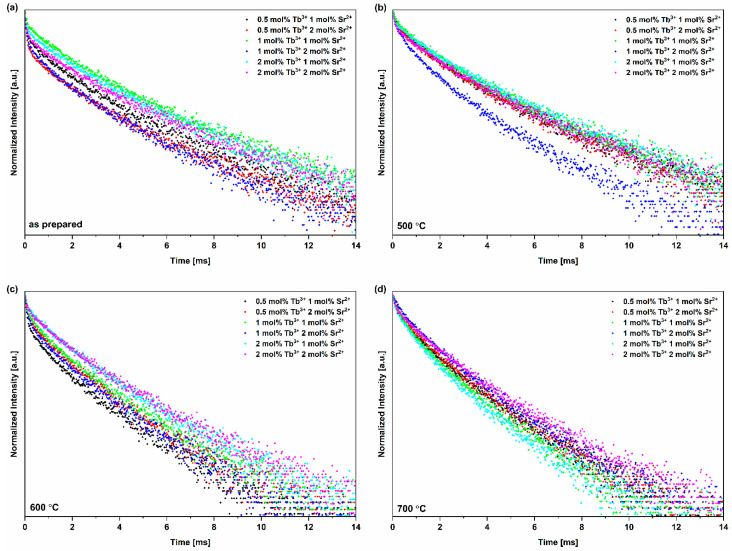
Fluorescence decay time of series (**a**) thermally untreated (as prepared), (**b**) annealed at 500 °C, (**c**) annealed at 600 °C and (**d**) annealed at 700 °C at an excitation wavelength of 266 nm and an observation wavelength of 545 nm (^5^D_4_→^7^F_5_ transition). For all obtained materials, the fluorescence decay is not monoexponential.

**Figure 8 nanomaterials-13-00457-f008:**
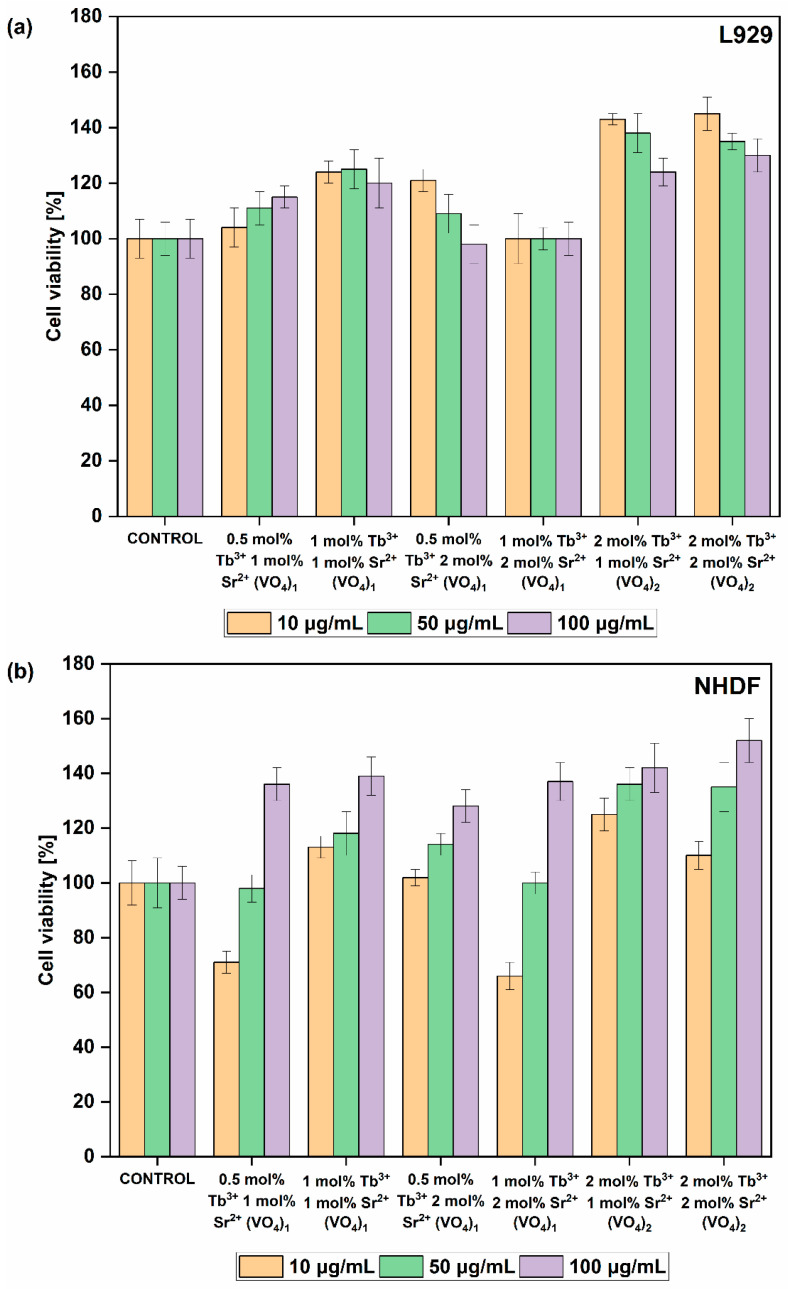
MTT cytotoxicity assay of phosphate–vanadate hydroxyapatite co-doped with Tb^3+^ and Sr^2+^ ions. The final concentrations of the tested compounds were established at 10 µg/mL, 50 µg/mL and 100 µg/mL. Cell lines used in the experiment were (**a**) L929 and (**b**) NHDF cells.

**Table 1 nanomaterials-13-00457-t001:** The elemental contents in the obtained materials (annealed at 600 °C) measured with the ICP-OES technique.

Material	n Ca (mol)	n Tb (mol)	n Sr (mol)	n P (mol)	n V (mol)
**Ca_9.85_Sr_0.1_Tb_0.05_(PO_4_)_5_(VO_4_)_1_(OH)_2_**	9.85	0.07	0.08	5.34	1.17
**Ca_9.75_Sr_0.2_Tb_0.05_(PO_4_)_5_(VO_4_)_1_(OH)_2_**	9.78	0.07	0.15	5.26	1.34
**Ca_9.8_Sr_0.1_Tb_0.1_(PO_4_)_5_(VO_4_)_1_(OH)_2_**	9.82	0.14	0.05	5.11	1.41
**Ca_9.7_Sr_0.2_Tb_0.1_(PO_4_)_5_(VO_4_)_1_(OH)_2_**	9.69	0.15	0.16	5.72	1.13
**Ca_9.7_Sr_0.1_Tb_0.2_(PO_4_)_4_(VO_4_)_2_(OH)_2_**	9.68	0.24	0.08	4.54	2.00
**Ca_9.6_Sr_0.2_Tb_0.2_(PO_4_)_4_(VO_4_)_2_(OH)_2_**	9.60	0.23	0.17	4.35	2.10

**Table 2 nanomaterials-13-00457-t002:** Band frequencies and assignment for the ATR-FT-IR spectra of HVp co-doped with 1 mol% Sr^2+^ and 1 mol% Tb^3+^ ions: samples thermally untreated (as prepared) and annealed at 500, 600 and 700 °C.

Frequencies (cm^−1^)				Mode
As Prepared	500 °C	600 °C	700 °C	
**470**	468	469	-	*ν_2_*(PO_4_^3−^) *ν_2_*(VO_4_^3−^) ν_4_(VO_4_^3−^)
**560**	563	560	562	ν_4_(PO_4_^3−^)
**601**	601	600	600	ν_4_(PO_4_^3−^)
**631**	632	630	633	ν_t_(OH^−^)
**850**	850	849	850	ν_3_(VO_4_^3−^)
**869**	869	870	871	ν_3_(VO_4_^3−^)
**887**	888	887	888	ν_3_(VO_4_^3−^)
**962**	962	960	962	ν_1_(PO_4_^3−^)
**1023**	1024	1024	1026	ν_3_(PO_4_^3−^)
**1086**	1088	1086	1087	ν_3_(PO_4_^3−^)
**3566**	3566	3567	3566	ν_s_(OH^−^)

**Table 3 nanomaterials-13-00457-t003:** The average fluorescence decay time for obtained materials measured in milliseconds.

Material	Decay Time (ms)			
	As Prepared	500 °C	600 °C	700 °C
Ca_9.85_Sr_0.1_Tb_0.05_(PO_4_)_5_(VO_4_)_1_(OH)_2_	2.42	2.47	1.61	1.69
Ca_9.75_Sr_0.2_Tb_0.05_(PO_4_)_5_(VO_4_)_1_(OH)_2_	2.40	2.50	1.73	1.64
Ca_9.8_Sr_0.1_Tb_0.1_(PO_4_)_5_(VO_4_)_1_(OH)_2_	2.71	2.59	1.82	1.49
Ca_9.7_Sr_0.2_Tb_0.1_(PO_4_)_5_(VO_4_)_1_(OH)_2_	2.17	1.79	1.70	1.66
Ca_9.7_Sr_0.1_Tb_0.2_(PO_4_)_4_(VO_4_)_2_(OH)_2_	2.69	2.66	1.97	1.41
Ca_9.6_Sr_0.2_Tb_0.2_(PO_4_)_4_(VO_4_)_2_(OH)_2_	2.77	2.63	2.10	1.93

## Data Availability

Data are available from the authors upon request.

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
