# Peer review of "Synthesis and Investigation of Physicochemical Properties and Biocompatibility of Phosphate–Vanadate Hydroxyapatite Co-Doped with Tb3+ and Sr2+ Ions"

_nanomaterials, 2023, doi:10.3390/nano13030457_

Round 1

Reviewer 1 Report

The paper's figures are unreadable and suspect the pdf creation process has mis-fired by creating blurry images. 

The English of the paper is rather poor and will cause confusion to readers.

The abstract says almost nothing and just states what has been done and not what was found

English language needs to be improved by a commercial editing service

There appears to be evidence for the substitutions (EDX map and ICP-OES) but the scientific explanations are convoluted and not readily understood which is compounded by the poor English. Why not use KBr disk IR spectra for better quality than ATR? Explain why this has not been considered?

It seems like broadenings in the XRD are all that is used to confirm substitution ...what about cell parameters ?  Needs clearer explanation for confirming substitution

Author Response

Dear Editor,

We would like to express our sincerest gratitude to the reviewers for their thoughtful evaluation of our manuscript. We have considered all raised questions, and carefully revised the manuscript according to the reviewers’ comments.  Moreover, all changes we have made to the original manuscript are marked with red colour in the text.

Reviewer 1:

Question 1: The paper's figures are unreadable and suspect the pdf creation process has mis-fired by creating blurry images.

Answer: Thank you for pointing this out, it has been corrected in the text.

Question 2: The English of the paper is rather poor and will cause confusion to readers.

Answer: Thank you for pointing this out, it has been corrected.

Question 3: The abstract says almost nothing and just states what has been done and not what was found.

Answer: Thank you for pointing that out. It has been changed to be more consistent.

Question 4: English language needs to be improved by a commercial editing service.

Answer: Thank you for pointing this out. It has been improved.

Question 5: There appears to be evidence for the substitutions (EDX map and ICP-OES) but the scientific explanations are convoluted and not readily understood which is compounded by the poor English. Why not use KBr disk IR spectra for better quality than ATR? Explain why this has not been considered?

Answer: In our previous papers we used ATR spectra because it provides enough data to determine presence of phosphate, vanadate and hydroxyl groups.

Question 5: It seems like broadenings in the XRD are all that is used to confirm substitution ...what about cell parameters?  Needs clearer explanation for confirming substitution

Answer: The parameters have been calculated but only for the pure-phase materials. We decided that in this case, giving these structural parameters it would be confused for readers. Since it is difficult to compare these calculated parameters between pure-phase and multiphase materials.

Reviewer 2 Report

In this paper, new, phosphate-vanadate hydroxyapatites co-doped with Sr2 and Tb3+ ions are presented. This is as a potential bioimaging fluorescent probe for regenerative medicine. It has certain practical value and research significance.

However, there are some questions about the research ideas and methods:

(1) This paper mainly discusses the influence of heat treatment temperature on strontium and vanadium codoped materials. All the data figures are developed around strontium and vanadium codoped materials, and the data of undoped samples are not given. Doping is modification rather than modification of the structure, and data graphs of undoped samples, such as XRD, etc., should also be given for comparison.

(2) The author can determine the optimal heat treatment temperature and doping concentration based on the structure or a certain property (such as biological activity, which was not tested in this paper, no matter how good the performance after doping is, if it does not have biological activity, it will have no application value), rather than testing the performance of all the doped samples.

(3) In the paper, none of the figures are very clear.

(4) In the Section 3.2, adding a Schematic energy level diagram.

(5) In the Section 3.3, why the cell viability of high concentration treated samples is higher?

Author Response

Dear Editor,

We would like to express our sincerest gratitude to the reviewers for their thoughtful evaluation of our manuscript. We have considered all raised questions, and carefully revised the manuscript according to the reviewers’ comments.  Moreover, all changes we have made to the original manuscript are marked with red colour in the text.

Reviewer 2:

In this paper, new, phosphate-vanadate hydroxyapatites co-doped with Sr2 and Tb3+ ions are presented. This is as a potential bioimaging fluorescent probe for regenerative medicine. It has certain practical value and research significance.

However, there are some questions about the research ideas and methods:

Question 1: This paper mainly discusses the influence of heat treatment temperature on strontium and vanadium codoped materials. All the data figures are developed around strontium and vanadium codoped materials, and the data of undoped samples are not given. Doping is modification rather than modification of the structure, and data graphs of undoped samples, such as XRD, etc., should also be given for comparison.

Answer: Thank you for pointing this out. It has been added the ICSD database pattern (ICSD 253617) of HAp to compare the obtained materials with the hydroxyapatite with hexagonal space group. Furthermore, it has been chosen the dopants’ concentrations besing on our previous research. In this case, the luminescent properties of undoped materials were not studied  because the aim of the research was realted to obtain a luminescent probe exhibiting biological properties and to specify its properties.

Question 2: The author can determine the optimal heat treatment temperature and doping concentration based on the structure or a certain property (such as biological activity, which was not tested in this paper, no matter how good the performance after doping is, if it does not have biological activity, it will have no application value), rather than testing the performance of all the doped samples.

Answer: It has been chosen a temaperature at 600ËšC in accordance with patterns’ observation. Moreover, at this temperature dopants are stable in agreement with the theoretical values. Moreover, it has been analyzed by ICP-OES technique. Finally, it has been tested by biological properties all materials that were heated at 600oC because showing the best candidate for bio imaging probe (please see the Sections 4.).

Question 3: In the paper, none of the figures are very clear.

Answer: Thank you for pointing this out, we fixed it.

Question 4: In the Section 3.2, adding a Schematic energy level diagram.

Answer: A schematic energy level diagram for Eu3+ ions is well know, and published by us previously for similar materials therefore it has been decided to not add it in our paper.

Question 5: In the Section 3.3, why the cell viability of high concentration treated samples is higher?

Answer: The Cell viability of both used cell lines (L929 and NHDF) reaches the highest values in the highest tested concentration because the incorporated dopants especially Sr2+ ions and (VO43-) group exhibit positive biological activity [1-6]. Moreover, the combination of such elements especially in hydroxyapatite lattice that is also known as highly biocompatible material, even synthetic one, can promote cell proliferation [7,8].   

  1. Hamdan Alkhraisat, M., Moseke, C., Blanco, L., Barralet, J. E., Lopez-Carbacos, E., & Gbureck, U. (2008). Strontium modified biocements with zero order release kinetics. Biomaterials, 29(35), 4691–4697. https://doi.org/10.1016/j.biomaterials.2008.08.026
  2. Huang, M., Li, T., Pan, T., Zhao, N., Yao, Y., Zhai, Z., Zhou, J., Du, C., & Wang, Y. (2016). Controlling the strontium-doping in calcium phosphate microcapsules through yeastregulated biomimetic mineralization. Regenerative Biomaterials, 3(5), 269–276. https://doi.org/10.1093/rb/rbw025
  3. Ressler, A., Cvetnić, M., Antunović, M., Marijanović, I., Ivanković, M., & Ivanković, H. (2020). Strontium substituted biomimetic calcium phosphate system derived from cuttlefish bone. Journal of Biomedical Materials Research - Part B Applied Biomaterials, 108(4), 1697–1709. https://doi.org/10.1002/jbm.b.34515
  4. A Morita, J Zhu, N Suzuki, A Enomoto, Y Matsumoto, M Tomita, T Suzuki, K Ohtomo, & Y Hosoi. (2006). Sodium orthovanadate suppresses DNA damage-induced caspase activation and apoptosis by inactivating p53. Cell Death Differ., 13(3), 499–511. https://doi.org/10.1038/sj.cdd.4401768
  5. Wang B, Tanaka K, Morita A, Ninomiya Y, Maruyama K, Fujita K, Hosoi Y, & Nenoi M. (2013). Sodium orthovanadate (vanadate), a potent mitigator of radiation-induced damage to the hematopoietic system in mice. J Radiat Res., 54(4), 620–629. https://doi.org/10.1093/jrr/rrs140
  6. Akinori Morita, Shinichi Yamamoto, Bing Wang, Kaoru Tanaka, Norio Suzuki, Shin Aoki, Azusa Ito, Tomohisa Nanao, Soichiro Ohya, Minako Yoshino, Jin Zhu, Atsushi Enomoto, Yoshihisa Matsumoto, Osamu Funatsu, Yoshio Hosoi, & Masahiko Ikekita. (2010). Sodium orthovanadate inhibits p53-mediated apoptosis. Cancer Research, 70(1), 257–265. https://doi.org/10.1158/0008-5472.CAN-08-3771
  7. Fiume, E., Magnaterra, G., Rahdar, A., Verné, E., & Baino, F. (2021). Hydroxyapatite for biomedical applications: A short overview. In Ceramics (Vol. 4, Issue 4, pp. 542–563). https://doi.org/10.3390/ceramics4040039
  8. Villa, M. M., Wang, L., Huang, J., Rowe, D. W., & Wei, M. (2015). Bone tissue engineering with a collagen-hydroxyapatite scaffold and culture expanded bone marrow stromal cells. Journal of Biomedical Materials Research - Part B Applied Biomaterials, 103(2), 243–253. https://doi.org/10.1002/jbm.b.33225

Reviewer 3 Report

I think the authors should put more attention to detail. In general, it can be interesting but there are changes to be taken:

The abstract is too sloppy.

The introduction and materials and methods sections need to be more accurately described. It is redundant and boring writing. 

Author Response

Dear Editor,

We would like to express our sincerest gratitude to the reviewers for their thoughtful evaluation of our manuscript. We have considered all raised questions, and carefully revised the manuscript according to the reviewers’ comments.  Moreover, all changes we have made to the original manuscript are marked with red colour in the text.

Reviewer 3:

I think the authors should put more attention to detail. In general, it can be interesting but there are changes to be taken:

Question 1: The abstract is too sloppy.

Answer: Thank you for pointing that out. It has been changed to be more consistent.

Question 1: The introduction and materials and methods sections need to be more accurately described. It is redundant and boring writing.

Answer: The extensive introduction was a result of an explanation related to our choices and perspectives of the dopants in the hydroxyapatite compounds.

Round 2

Reviewer 2 Report

1. The title below the figure should be consistent with the label on the figure. The title below Figure 2.2 is A, not a.

2. Note the superscripts in the references.

3. Where are the experiments on biological activity?

Author Response

Dear Editor,

We would like to express our sincerest gratitude to the reviewers for their thoughtful evaluation of our manuscript. We have considered all raised questions, and carefully revised the manuscript according to the reviewers’ comments.  Moreover, all changes we have made to the original manuscript are marked with red colour in the text.

Reviewer 2:

Comments and Suggestions for Authors

Question 1: The title below the figure should be consistent with the label on the figure. The title below Figure 2.2 is A, not a.

Answer: Thank you for pointing this out, it has been changed.

Question 2: Note the superscripts in the references.

Answer: Likewise, it has been changed.

Question 3:  Where are the experiments on biological activity?

Answer: We discussed biological properties of the obtained materials in sections 2.4 (Materials and methods) and further investigated it in section 3.3 (Evaluation of biological properties).

Reviewer 3 Report

No further comments.

Author Response

Dear Editor,

We would like to express our sincerest gratitude to the reviewers for their thoughtful evaluation of our manuscript. We have considered all raised questions, and carefully revised the manuscript according to the reviewers’ comments. 

Reviewer 3:

Comments and Suggestions for Authors

No further comments.

Answer:

Thank you for revising our paper again and positive suggestions.
